# Synthesis and Biological Activity of Chromeno[3,2-*c*]Pyridines

**DOI:** 10.3390/molecules29214997

**Published:** 2024-10-22

**Authors:** Anna V. Listratova, Roman S. Borisov, Nikolay Yu. Polovkov, Larisa N. Kulikova

**Affiliations:** 1Organic Chemistry Department, Peoples’ Friendship University of Russia Named after Patrice Lumumba (RUDN University), 6 Miklukho-Maklaya St., 117198 Moscow, Russia; listratova-av@rudn.ru; 2A.V.Topchiev Institute of Petrochemical Synthesis RAS, 29 Leninsky Prospekt, 119991 Moscow, Russia; borisov@ips.ac.ru (R.S.B.); polovkov@ips.ac.ru (N.Y.P.)

**Keywords:** chromenopyridines, chromeno[3,2-*c*]pyridines, chromeno[3,2-*c*]quinolines, synthesis, biological activities

## Abstract

The review summarizes all synthetic methodologies for the preparation of chromeno[3,2-*c*]pyridines and chromeno[3,2-*c*]quinolines. The proposed approaches are systemized based on ways for the construction of the heterocyclic system. The presence of these compounds in nature and their bioactivity are also discussed. Natural products with an annelated chromeno[3,2-*c*]pyridine fragment are well-known and a number of alkaloids derived from this system as a key core have been recently isolated. These compounds demonstrate antimicrobial, antivirus, and cytotoxic activities, making chromeno[3,2-*c*]pyridine structural motifs promising for medicinal chemistry.

## 1. Introduction

Quinoline, pyridine, chromene, and especially its derivatives chromones, represent privileged scaffolds and occupy a unique place in the field of synthetic and medicinal chemistry due to their wide spectrum of biological activities [1,2,3,4,5,6,7,8,9,10,11,12]. The combination of these heterocyclic systems in a single molecule promises to open up new opportunities for finding desirable properties and thus encourages the scientific community to develop methods for constructing such fused hybrids. Indeed, chromenopyridines are a structurally diverse class of compounds with a wide range of biological activities that are increasing in presence in pharmaceuticals. Moreover, studies show that they can be applied as dyestuffs [13,14,15], luminescence intensifiers [16], and fluorescent dyes [17]. An analysis of data from the literature has revealed that for over the past three decades, a number of reviews on chromenopyridines have been published [18,19,20], but to our surprise, chromeno[3,2-*c*]pyridines and chromeno[3,2-*c*]quinolines have not been completely covered; so far, only scattered examples have been described in some of them [21,22]. Here, we try to summarize all synthetic methodologies and present the results of our survey.

## 2. Recently Isolated Natural Chromeno[3,2-*c*]Pyridines and Their Bioactivities

Natural products with an annelated chromeno[3,2-*c*]pyridine fragment are well-known and were discovered long ago [23,24,25,26,27,28,29,30,31,32]. However, surprisingly, alkaloids derived from the chromeno[3,2-*c*]pyridine system as a key core were not been isolated until the twenty-first century. Only in recent decades have papers devoted to the isolation and identification of chromeno[3,2-*c*]pyridines alkaloid begun to appear. Thus, in 2002, Paune et al. disclosed and characterized a novel series of tricyclic natural product-derived metallo-*β*-lactamase inhibitors. Among three natural products isolated from the fungus *Chaetomium funicola*, there was an alkaloid with the unique chromeno[3,2-*c*]pyridine nucleus. The first representative of the new class was labeled as SB236049. The biological test of SB236049 revealed inhibitory activity towards the *Bacillus cereus II* and *Bacteroides fragilis* CfiA metallo-*β*-lactamases [33] (Figure 1).

Later in 2013, Gan et al. succeeded in isolating another alkaloid with the chromeno[3,2-*c*]pyridine core–7-hydroxy-8-methoxy-3-methyl-10-oxo-10*H*-chromeno[3,2-*c*]-pyridine-9-carboxylic acid **1** from the cultures of *Penicillium* sp. I09F 484 (Figure 1) [34]. The alkaloid displayed inhibitory activity against New Delhi metallo-*β*-lactamase 1 with IC_50_ values of 87.9 μM, but it did not reveal antimicrobial, antiviral, and cytotoxic activities at a concentration of 10 μM. The authors also proposed the biosynthetic pathway, suggesting that the chromeno[3,2-*c*]pyridine moiety was generated by a common biosynthesis strategy from a heptaketide precursor (Figure 1).

In 2017, when analyzing metabolites from the mangrove endophytic fungus *Diaporthe phaseolorum* SKS019 obtained from holy mangrove (*Acanthus ilicifilius*), Cui et al. separated and identified four previously unknown alkaloids of this type. The new chromeno[3,2-*c*]pyridine alkaloids, labeled as Diaporphasines A–D, were tested on five tumor cell lines, but unfortunately, they exhibited no significant inhibitory activity [35]. More recently, the Diaporphasine series has been extended, with Diaporphasine E isolated by Phutthacharoen et al. from a mycelial extract of a saprotrophic fungus *Lachnum* sp. IW157 growing on the common reed grass *Phragmites communis* (Figure 1) [36]. Diaporphasines E revealed potent cytotoxicity against the tested cell lines L929 and KB3.1 with IC_50_ values of 0.9 and 3.7 μM.

Chromeno[3,2-*c*]pyridine, named Phomochromenones C (Figure 2), was isolated from metabolites of the endophytic fungus *Phomopsis* sp. HNY29-2B, which was derived from the mangrove plant *Acanthus ilicifolius* Linn [37]. Later Phomochromenones C was also separated from the culture of *Phomopsis asparagi* DHS-48 from the Chinese mangrove *Rhizophora mangle* [38] and the marine-derived fungus *Diaporthe* sp. XW12–1 [39].

In 2021–2022, scientific groups from China published a series of papers devoted to isolation and identification of chromenopyridine alkaloids from different species of *Thalictrum*. Thus, Yang et al. reported on the isolation of three previously unknown chromeno[3,2-*c*]pyridine alkaloids **2**–**4** from the whole plants of *Thalictrum scabrifolium*. The compounds were tested and showed antirotavirus activity (Figure 3, line 1) [40]. Later, and from the same species, Yin et al. managed to obtain two more natural chromeno[3,2-*c*]pyridines **5**–**6**, which possessed high antibacterial activity against 12 microbial strains isolated from the saliva of smokers (Figure 3, line 2) [41].

Chromeno[3,2-*c*]pyridines **7**–**8**, isolated by Hu et al. from the whole plants of *Thalictrum finetii*, exhibited high antirotavirus activity (Figure 3, line 3) [42].

Exploring the whole plants of *Thalictrum microgynum*, Wu et al. succeeded in isolating two new alkaloids **9**–**10**, comprising chromeno[3,2-*c*]pyridine skeleton in their structures. The obtained chromeno[3,2-*c*]pyridines showed anti-tobacco mosaic virus (anti-TMV) activity [43] (Figure 3, line 4).

In 2023, Wu et al., screening metabolites obtained from the cigar-tobacco-leaf-derived endophytic fungi *Aspergillus lentulus*, isolated two known chromeno[3,2-*c*]pyridine alkaloids (Figure 3, compound **4** and compound **9**). The anti-TMV activities test of compound **4** revealed its high anti-TMV activities with inhibition rate of 38.5% [44].

## 3. Synthetic Ways to Chromeno[3,2-*c*]Pyridines and Chromeno[3,2-*c*]Quinolines

### 3.1. Construction of Chromene Fragment

#### 3.1.1. Synthesis Based on Cyclization of Pyridyl(Quinolyl) Phenyl Ethers

The first suggested synthetic ways towards chromeno[3,2-*c*]pyridines were based on the cyclization of pyridyl phenyl ethers decorated with appropriate groups—cyano, ester, carboxylic acid, etc. In 1955, Kruger et al. realized the synthesis of chromeno[3,2-*c*]pyridine **14** via the cyclization of cyanopyridyl phenyl ether [45]. A four-step sequence commenced with a condensation of 4-chloro-3-nitropyridine with phenol to give intermediate 3-nitropyridyl phenyl ether **11**. Reduction of the nitro-group of **11** to amine **12**, followed by diazotation and cyanation, afforded cyanopyridyl phenyl ether **13**. The final cyclization of it under the action of H_2_SO_4_ at 195 °C led to the target chromeno[3,2-*c*]pyridine **14** in 10% yield (Figure 2).

In 1970, Bloomfield et al. showed that ester decorated phenyl pyridyl ether were also appropriate for furnishing the chromene fragment and realized a successful attempt at cyclization of ether **16** [46]. Obtained by treatment of the 4-chloroquinolines **15** with phenol at 160 °C, phenyl ethers **16** readily cyclized in hot polyphosphoric acid to give chromeno[3,2-*c*]quinolines **17** in 58–82% yield (Figure 3).

Later, in 1975, employing 4-phenoxynicotinic acid **20** as a substrate, Villani et al. succeeded in achieving chromeno[3,2-*c*]pyridine **14** in 91% yield through the cyclization stage, which was carried out under the action of PPA (Figure 4) [47]. The synthesis of the desired 4-phenoxynicotinic acid required four steps and included the interaction of the starting 4-nitro-3-picoline 1-oxide and phenol, resulting in the formation of ether **18,** which was *N*-deoxygenated to give pyridyl ether **19**. Oxidation of the latter with aqueous KMnO_4_ led to the desired substrate **20**.

In 1999, Khodair et al. also exploited this strategy [48]. Phenyl pyridyl ethers **23**, derived from 4-chloro-3-nitro substituted quinolones **21** and corresponding salicylic acids or salicylaldehydes **22**, underwent cyclization in benzene under reflux to produce chromeno[3,2-*c*]quinolines **24** (Figure 5). It was also shown that the cyclization could be realized as a one-pot process.

In their research, Okada et al. took 3-trifluoroacetyl-4-quinolyl ethers **26** as a substrate for cyclization to obtain chromeno[3,2-*c*]quinolones **27** bearing a trifluoromethyl group in high yields. The required precursors **26** were synthesized by a two-step sequence including trifluoroacylation of 4-dimethylaminoquinoline followed by an exchange reaction with phenols **25**. The cyclization of compound **26** with trifluoromethansulfonic acid (TFSA) proceeded smoothly at room temperature to give the final product **27** (Figure 6) [49].

In 2017, Hong et al. suggested an efficient method to assemble the chromeno[3,2-*c*]quinoline core based on a one-pot oxidative *O*-arylation, Pd^0^-catalyzed C(sp^3^)-H arylation sequence [50]. In situ, generated from arylols **28** and trivalentaryl iodine reagents **29**, without isolation ethers **30** underwent Pd^0^-catalyzed C(sp^3^)-H arylation to give the target products **31** in good yields (54–80%) (Figure 7). The cyclization step occurred exclusively at the sterically less hindered position when methyl and ethyl groups were both present on the aryl ring of the iodine reagent, and tolerated both electron-donating and electron-withdrawing groups on the 6-position of the quinoline ring.

Recently, Kardile et al. returned to the idea of the cyclization of pyridyl phenyl ether having a carboxylic acid group and performed a six-step synthesis of chromeno[3,2-*c*]quinolines **33**, starting from 4-bromoaniline and malonic acid (Figure 8). The final cyclization step of ether **32** was carried out under trifluoroacetic anhydride in a mixture of THF/DCM (1:1) at 0 °C—room temperature to afford the target compound **33** in 81% yield. Further, compound **33** produced a series of chromeno[3,2-*c*]quinolines **34** by reaction with Grignard reagent [51].

Chen et al. also resorted to the cyclization of pyridyl phenyl ether as a methodology of the synthesis of chromeno[3,2-*c*]pyridine **39** necessary for their research on the discovery of novel chronic hepatitis B virus cccDNA reducers [52]. The target chromeno[3,2-*c*]pyridine **39** was obtained through Cu-catalyzed O-arylation between methyl 4,6-dicholonicotinate **36**, derived from the corresponding acid **35**, and 2-fluorophenol followed by the hydrolysis of intermediate ester **37**, and the cyclization of the resultant ether **38** under the action of sulfuric acid at 100 °C (Figure 9). The successive substitution of chlorine atom with pyrrolidine and K_3_PO_4_ as a base gave compound **40**.

#### 3.1.2. Synthesis Based on Morpholine Enamine

One of the most promising and simple synthetic pathways towards chromeno[3,2-*c*]pyridines is a condensation of heterocyclic morpholine enamines with salicylaldehydes. This approach was first proposed by Sliwa et al. in 1977 [53]. A condensation of morpholine enamine with salicylaldehyde in boiling hexane followed by the direct oxidation of intermediate **41** with chromium trioxide-pyridine afforded chromeno[3,2-*c*]pyridine **42** in 35% yield. When refluxed in xylene in the presence of palladium-charcoal, compound **42** underwent debenzylation and aromatization to give **14**, which was reduced by lithium aluminum hydride to 10*H*-chromeno[3,2-*c*]pyridine **43** (Figure 10).

Ten years later, the authors expanded their idea and showed that condensation could be realized between heterocyclic morphine enamine **44** and *β*-ketoester **45** when heated in xylene. In addition, the resulting octahydro derivative **46** could undergo partial or complete aromatization, leading to either chromeno[3,2-*c*]pyridine **47** or chromeno[3,2-*c*]pyridine **14** (Figure 11) [54].

Later, the above-mentioned methodology was successfully applied by other scientific groups to create a series of differently substituted chromeno[3,2-*c*]pyridines **51–52** (Figure 12) [55,56,57,58]. It was shown that the reaction features simplicity of performance and is tolerant to both electron-donating and electron-withdrawing groups. Moreover, it was found that when heated in o-xylene in the presence of TsOH, intermediate **50** was converted into chromeno[3,2-*c*]pyridines in moderate yields. Some of the obtained chromeno[3,2-*c*]pyridines **51** were chemically modified to increase the diversity of the class for biological tests [56,57,58]. Several derivatives showed multifaceted profiles of promising anti-Alzheimer’s disease properties and well-balanced multitarget inhibitory activity. Inhibitory activities against monoamine oxidase A and B (MAO A and B), acetyl- and butyrylcholinesterase (AChE and BChE), and anti-aggregation activity against β-amyloid were demonstrated. One of the compounds, a potent and selective inhibitor of human MAO B (IC50 = 0.89 μM), was shown to be a safe neuroprotector in a human neuroblastoma cell line (SH-SY5Y), improving viability impaired by Aβ1–42 and prooxidant damage.

#### 3.1.3. Synthesis Based on Intramolecular Cyclization via Nucleophilic Substitution of Halogen

In 1971, Harnisch used an electrophilic substitution/nucleophilic substitution sequence of reactions for the synthesis of dye **57** possessing the chromeno[3,2-*c*]pyridine fragment [14]. The starting 4-chloroquinoline-3-caraldehyde **55** and *N*,*N*-dimethyl-3-aminophenol **56** underwent cyclization in glacial acetic acid under reflux to give dye **57** in 80% (Figure 13). Later, the same principle for the synthesis of dyes and fluorescent markers was applied by other scientific groups [17].

In 1988, Marsais et al. also employed nucleophilic substitution of halogen atom as the final step for the construction of chromeno[3,2-*c*]pyridine synthon. To activate the halogen atom and facilitate the nucleophilic substitution, 4-fluoropyridine was submitted to metalation with LDA and *o*-anisaldehyde to give secondary alcohol **58**, which when oxidized by MnO_2_ afforded 4-fluoropyridine **59**, activated by 3-carbonyl moiety. Intramolecular cyclization of the latter occurred in boiling pyridinium chloride, providing chromeno[3,2-*c*]pyridine **14** in 80% yield (Figure 14) [59].

While exploring the synthesis of azafluorenones through Pd-catalyzed intramolecular arylation of diarylketones bearing a halogen at the 2-position in one of the aryl groups, Marquise et al. observed that under the suggested conditions (2-chlorophenyl)(2-methoxypyridin-3-yl)methanone **60** preferably underwent an intramolecular cyclization, resulting in the formation of chromeno[3,2-*c*]pyridine **61**, which was the only product of the reaction (Figure 15) [60].

#### 3.1.4. Synthesis Based on Chromene-3-Thiocarboxamide

In 2001, El-Sayed obtained a series of chromeno[3,2-*c*]pyridines **62** from chromene-3-thiocarboxamides **61**. The substrates **61** reacted with oxalyl chloride in the presence of triethylamine in dioxane at room temperature to give chromeno[3,2-*c*]pyridines **62** in good yields, whereas the conversion of chromene **61** into compound **64** required more rigid conditions and proceeded in two steps. Alkylation of thiocarboxamide fragment with bromomalononitrile and the sequent intramolecular cyclization of the resultant chromene **63** in refluxing DMSO afforded chromeno[3,2-*c*]pyridine **64** in 67% yield (Figure 16) [61].

In a later paper, it was demonstrated that the interaction of 2-imino-chromen-3-thiocarboxamide **65** with malononitrile in EtOH in the presence of piperidine went through a cascade of reactions leading to the final compound **67** in 64% yield (Figure 17). The process occurred at room temperature, but when the temperature was raised to boiling EtOH, another pathway was preferable, resulting in chromeno[2,3-*b*]pyridine [62].

#### 3.1.5. Synthesis Based on Knoevenagel Condensation/Michael Addition Sequence

In 2015, Kumar et al. reported an efficient synthesis of a series of novel chromeno[3,2-*c*]pyridines **72** via Michael addition/intramolecular *O*-cyclization/elimination cascade [63]. 3,5-((*E*)-arylidene)-1-alkylpiperidin-4-ones **70**, derived by Knoevenagel condensation of two molecules of aryl aldehyde **69** and *N*-alkyl(benzyl)piperidone **68**, reacted with cyclic 1,3-diketones **71** in acetic acid under reflux to afford chromeno[3,2-*c*]pyridines **72** in excellent yields (Figure 18). The process tolerates aldehydes bearing electron-withdrawing or electron-donating groups in the aryl rings. It is worth noting that the attempt to find conditions suitable to carry out a three-component reaction starting from the corresponding piperidones, aryl aldehydes, and cyclohexane-1,3-diones failed; instead of the targeted chromenopyridine derivatives, xanthene was formed.

In 2016, while working on a three-component strategy towards dibenzo[*b*,*h*][1,6]naphthyridine from aryl aldehydes **73**, 3-(arylamino)cyclohex-2-enone **74** and 4-hydroxyquinolinone **75**, Wang et al. developed an efficient method for the synthesis of chromeno[3,2-*c*]quinolones **77** [64]. It was observed that a three-component reaction of the starting compounds **73**, **74,** and **75**, catalyzed by TsOH and heated to 140 °C in ionic liquid, provided chromeno[3,2-*c*]quinolones **77** in high yields. The temperature value and the presence of TsOH appeared to be crucial; without them, the final cyclization did not occur. As it was in the previous example, the reaction starts with the Knoevenagel condensation of aryl aldehyde **73** and 4-hydroxyquinolinone **75**; the resulting quinolone-2,4-dione **76** undergoes consecutive Michael addition, protonation, secondary Michael addition, and deamination (Figure 19). Interestingly, again, the idea to use dimedone instead of 3-(arylamino)cyclohex-2-enone **74** in a three-component reaction failed.

Recently Kamali et al. managed to select conditions and starting material suitable for a three-component reaction with dimedone and proposed a facile, one-pot three-component synthesis towards chromeno[3,2-*c*]pyridine-1,9-diones **81** [65]. The chromeno[3,2-*c*]pyridine system was built up through a SnCl_2_·2H_2_O-mediated sequence of Knoevenagel condensation/Michael addition/Knoevenagel condensation form 4-hydroxypyridine-2-ones **78**, aldehydes **79**, and dimedone in ethanol at 70 °C (Figure 20). The mechanism of the transformation resembles the examples depicted in previous Figure 19. Despite all the advantages of the reaction (environmentally friendly, mild conditions, operational simplicity, high yields), it featured a drawback as it could only be realized for aromatic aldehydes.

The similar strategy based on Knoevenagel condensation/Michael addition sequence was successfully applied to the synthesis of benzo[5,6]chromeno[3,2-*c*]quinolones **83**. By taking arylglyoxal monohydrates **82**, quinoline-2,4-dione, and *β*-naphthol as starting materials, Orang et al. succeeded in developing a TsOH-catalyzed one-pot, three-component reaction [66,67]. The reaction occurred in a water/EtOH mixture (2:1) under reflux to give compounds **83** in 88–92% yields. It is believed that the process commences with the TsOH-catalyzed in situ generation of arylglyoxal, followed by Knoevenagel condensation with quinoline-2,4-dione. The key step of the construction of chromene fragment is Michael addition of *β*-naphthol, then the subsequent *O*-cyclization and the elimination furnish formation of benzo[5,6]chromeno[3,2-*c*]quinolones **83** (Figure 21).

#### 3.1.6. Miscellaneous

In 2022, Yang et al. published a paper on a methodology for the efficient construction of *β*-enamino diketones **86** through the DABCO-catalyzed, CH_3_NO_2_-mediated three-component reaction of 1,3-cyclodiketones **84**, furfurals **85**, and allylamine in reflux toluene. Further, the obtained compounds **86** were successfully converted to chromeno[3,2-*c*]quinolines **87** via bromination with NBS [68]. The transformation presumably commences with the bromination of the double bond on the epoxyisoindole of *β*-enamino diketones **86** with NBS to produce cyclic brominium ion **A**. The succinimide anion, resulting from the bromination, takes a proton from *β*-enamino nitrogen, followed by electron delocalization to give zwitterion **B**. The succeeding rotation of the C−C single bond between quinoline-2,4-dione and epoxyisoindole moieties of **B** favors the cyclic brominium fragment for the final intramolecular S_N_2 ring opening of the cyclicbromonium ion by the alkoxide anion to afford the cyclized product **87** (Figure 22). The authors also showed that compounds **87** could be diastereoselectively reduced by NaBH_4_ to chromeno[3,2-*c*]quinolones **88**.

### 3.2. Construction of Pyridine Fragment

#### 3.2.1. Synthesis Based on Ethyl Coumarin-3-Carboxylate

In 1980, Briet et al. released a patent describing the synthesis and antidepressant activity of a series of chromeno[3,2-*c*]pyridines **92**, among which the hit compound was lortalamine [69]. The chromeno[3,2-*c*]pyridine system was constructed in a two-step procedure from ethyl coumarin-3-carboxylates **89** and 1-alkyl-4-piperidones **90**. The Michael addition of *N*-substituted 4-piperidones **90** to coumarin derivatives **89** and the subsequent cleavage of the resultant adduct by ammonia, generated from ammonium acetate or primary amines **91,** occurred when the reaction mixtures were heated with or without an alcohol solvent. The following treatment with boiling concentrated hydrochloric acid caused cyclization to give the final product **92** (Figure 23). It was demonstrated that when the treatment was realized by cold concentrated hydrochloric acid *β*-ketoesters **93** were isolated, which successfully underwent decarboxylation when heated in sodium bicarbonate. Some *N*-benzyl substituted chromenopyridines **92** went through debenzylation to afford NH-chromonemopyridines. Later, it was shown that NH-chromonemopyridines could be obtained using *N*-Boc-4-piperidone [70].

Different groups of chemists undertook attempts to carry out the asymmetric synthesis of lortalamine or its analogs. Three papers appeared describing the syntheses of enantiomers [71,72,73]. All these syntheses were based on the methodology previously described in the patent.

#### 3.2.2. Synthesis Based on 3-Carbonylchromones

In 1988, Ghosh et al. synthesized several chromenopyridines from 2-(2-dimethylaminoethenyl)chromones **95** derived by methylenation of 2-methyl 3-carbonyl chromones **94** with *N*,*N*-dimethylformamide dimethyl acetal (DMFDA). When reacted with *N*-nucleophiles, chromones **95** underwent 1,6-addition-elimination sequence leading to enamines **96**; further, they were cyclized in acetic acid at the reflux to give chromeno[3,2-*c*]pyridines **97**, **98**, or **99**. Chromenopyridine *N*-acetylinide **97** was converted into compound **98** in boiling ethylene glycol with 70–80% yields (Figure 24) [74].

In another study, Ghosh et al. chose 3-carbonyl chromones **100** as the substrates for condensation with DMFDA. The obtained chromones **101** were submitted into reactions with *N*-nucleophiles in refluxing ethanol and, depending on the nature of the nucleophile, chromeno[3,2-*c*]pyridines **102** or chromeno[3,2-*c*]pyridinium salts **103** were formed. The formation of **102** was accompanied by phenyl pyridyl ketone **104** (Figure 25) [75].

In 1990, Rajagopal et al. accomplished the synthesis of fluorescent 2,3-fused couramin derivatives, including chromeno[3,2-*c*]pyridines **107** and **108** [76]. 3-Carboxamide derivative **105**, derived from 4-diethylaminosalicylaldehyde and cyanoacetamide, reacted with *p*-nitrobenzylcyanide or benzimidazo-2-acetonitrile **106** in DMF under reflux in the presence of pyridines as a catalyst, giving chromeno[3,2-*c*]pyridines **107** in 60% and 65% yield, respectively (Figure 26). When refluxed in DFA with esters or cyanoacetamide **108**, the starting coumarin **105** was converted into chromeno[3,2-*c*]pyridines **109** in 30–70% yields (Figure 26).

In 2008, Plaskon et al. described the synthesis of 7*H*-chromeno[3,2-*c*]quinoline-7-ones **111** employing a TMSCl-promoted cyclization of 3-formylchromone with various anilines **110** (Figure 27) [77]. On heating a solution of 3-formylchromone with anilines **110** in the presence of TMSCl in DMF at 100 °C, chromeno[3,2-*c*]quinolines **111** were obtained in 39–67% yields. The authors proved that the anilines **110** with substituents, which withdraw electrons from the *ortho*-position or increase electron density on the nitrogen-favored formation of chromeno[3,2-*c*]quinolones **111**; otherwise the cyclization did not occur.

In 2012, an unusual way for constructing a chromeno[3,2-*c*]pyridine framework was found by Wittstein et al., who were the first to demonstrate that conjugated *N*-phenyl-*C*-chromonyl nitrones **112** behaved as 1,5-dipoles. Synthesized by the condensation of 3-formyl chromones with phenylhydroxylamine, *N*-phenyl nitrones **112** reacted with zwitterionic allenoate **113**, in situ generated from triphenylphosphine and DMAD, to give chromeno[3,2-*c*]pyridines **114** (Figure 28) [78]. The transformation proceeds via [5+3] cycloaddition followed by rearrangement with the elimination of Ph_3_PO and 6*π*-electrocyclization to give the final compound **114**. It is noteworthy that *N*-substituents of nitrones **112** displayed an unusual control over the pathway of the transformations. Thus, *N*-alkyl substituted nitrones **112** with zwitterionic allenoate **113** provided an expected [3+2] cycloadducts instead of the formation of chromenopyridine synthon.

Another interesting way of building up chromeno[3,2-*c*]pyridine moiety starting from formylchromone was revealed by Bandyopadhyay et al. The Ugi adduct chromones **119**, synthesized from 3-formylchromone **115**, o-haloanilines **116**, isocyanides **117,** and carboxylic acids **118**, underwent a ligand-free Pd-catalyzed intramolecular *C*-arylation at the C-2 or C-3 position of the chromone with the *o*-halophenyl group to give compound **120** instead of C–N coupling between the *o*-halophenyl and the amide NH group. The reaction proceeded under an argon atmosphere with PdCl_2_ as a catalyst, KOAc, and DMF at 100–110 °C (Figure 29) [79].

#### 3.2.3. Synthesis Based on Diels-Alder Reaction

In their ongoing research focusing on the search for substances with antituberculous activity, Sriram et al. synthesized a series of chromeno[3,2-*c*]pyridines **124** using a one-pot two-step process. The first step involved a MW-promoted condensation of 3-formyl chromone with 2-amino-3-methylpyridine or 2-amino-3,5-dibromopyridine **121**, which was followed by MW-indium triflate-assisted hetero-Diels-Alder reaction of intermediate Schiff bases **122** and *N*-(prop-2-yn-1-yl)arylamides **123**. The target compounds were isolated in good yields (Figure 30) [80]. Some of these substances showed preliminary in-vitro and in vivo activity against *Mycobacterium tuberculosis* H37Rv (MTB) and multidrug-resistant *M. tuberculosis* (MDR-TB).

The possibility of exploiting the [4+2]-cycloaddition reactions of *o*-quinone methides with electron-rich olefines in the construction of the chromeno[3,2-*c*]pyridine system was demonstrated by Popova et al. [81]. The heating of 4-(1-methyl-1,2,3,6-tetrahydropyridin-4-yl)morpholine **126** with Mannich bases of isoflavones **125** in toluene, followed by the treatment with formic acid in isopropanol, led to the formation of pyrano[2′,3′:5,6]chromeno[3,2-*c*]pyridine **129** (Figure 31A). It is believed that the sequence of reactions begins with the generation of the *o*-quinone methide intermediate **127**, which undergoes the hetero-Diels–Alder reaction with 4-(1-methyl-1,2,3,6-tetrahydropyridin-4-yl)morpholine **126** to give unstable adducts **128**, the hydrolysis of which completes the transformation, giving the targeted system **129**. It is worth mentioning that a ring–chain tautomerism of the ketone and hemiketal forms was observed for the synthesized compounds **129**. The same idea was successfully applied for the synthesis of furo [2′,3′:5,6]chromeno[3,2-*c*]pyridin-3(2*H*)-one **131** from 6-hydroxy-7-dimethylaminomethylaurones **130** (Figure 31B).

#### 3.2.4. Miscellaneous

In 2004, Abdelkhalik et al. described the synthesis of thiophene-fused chromeno[3,2-*c*]pyridine **134** from benzo[*h*]thieno[3,4-*c*]chromene **132** [82]. The starting compound **132** reacted with DMAD in reflux xylene to produce chromeno[3,2-*c*]pyridine **134**, a product of addition and subsequent water elimination. The authors also showed that the same compound **134** (in 82% yield) could be obtained through the cyclization of thienopyridine **133**, derived from the interaction of benzo[*h*]theino [3,4-*c*]chromene **132** with DMAD in DMF at reflux, when it was heated in xylene (Figure 32).

In 2012, Yan et al. reported on a direct synthesis of chromeno[3,2-*c*]pyridines **138** via a domino three-component reaction [83]. Based on the fact that *N*-unsubstituted aryl aldimines would act as nucleophiles in the Michael addition reaction with 3-(1-alkynyl)chromones **135** and could cause further transformations, a new effective cascade reaction was developed. NH-aldimines **137**, in situ generated by condensation of the corresponding aldehydes **136** with ammonium acetate, reacted with 3-(1-alkynyl)chromones **135** to give chromeno[3,2-*c*]pyridines **138** in 21–85% yield. The process occurred in DMF at 100 °C and was tolerant to a wide range of aryl aldehydes **136** and formaldehyde, whereas the use of both enolizable and α,β-unsaturated aldehydes led to complicated results. The plausible mechanism suggested that the domino process begins with the Michael addition of aldimine **137** to 3-(1-alkynyl)chromones **135** to give pyrone **A**, the subsequent cleavage of pyrone ring produces intermediate **B**, which undergoes intermolecular cyclization followed by 6*π*-electrocyclization and dehydration to end with the formation of the final product **138** (Figure 33).

In 2014, Hamada et al. reported the efficient synthesis of fused heterocycles through an acid-promoted cascade cyclization process of aryl group-substituted propargyl alcohol derivatives **139** with a *p*-hydroxybenzylamine unit [84]. Using phenol derivatives **139** as substrates, they obtained chromeno[3,2-*c*]pyridine **140** in moderated yield. The cascade cyclization commences with an acid-promoted intramolecular *ipso*-Friedel-Crafts alkylation of phenol derivatives **139**; the subsequent rearomatization-promoted C–C bond cleavage gives an iminium cation **A**, which goes through aza-Prins reaction, and the final 6-membered ring cyclization of the resulting allyl cation **B** affords chromeno[3,2-*c*]pyridine product **140** (Figure 34).

Investigating the substrate scope of Pd(II)-catalyzed tandem C–H alkenylation/C–O cyclization reactions in *o*-hydroxyphenyl decorated flavone derivatives, Kim et al. demonstrated that the revealed process was flexible and could be successfully applied for *N*-sulfonyl derivative **141** [85]. Flavone **141** reacted with *n*-butyl acrylate at 120 °C in *t*-BuOH in the presence of Pd(acac)_2_ as a catalyst, Cs_2_CO_3_ as a base, and Al_2_O_3_ as an additive, to afford chromeno[3,2-*c*]quinoline **142** in 62% yield (Figure 35).

An interesting example of a one-pot synthesis of a novel series of chromenopyridodiazepinone **144** with chromeno[3,2-*c*]pyridine moiety was suggested by Bouchama et al. [86]. The synthesis was based on the nucleophilic addition of ethane-1,2-diamine to (*E*,*E*)-3-[3-(2-hydroxyphenyl)-3-oxoprop-1-en-1-yl]-2-styrylchromones **143** and proceeded at room temperature under mild conditions. Chromenopyridodiazepinone **144** resulted from a tandem process involving the Michael addition of one amine group of ethane-1,2-diamine, the subsequent intramolecular heterocyclization through a 1,6-conjugate addition, and the final imine condensation (Figure 36).

In 2019, Kumar et al. described an approach towards chromeno[3,2-*c*]quinolones **147** from 2-(2-aminophenyl)chromen-4-ones **145** through a Fe^III^-catalyzed imine formation/C-C coupling/oxidation cascade [87]. The starting substrates **145** interacted with aromatic aldehydes **146**, bearing both electron-donating and electron-withdrawing groups, in the presence of FeCl_3_ as a catalyst in nitrobenzene under an argon atmosphere (Figure 37). It is suggested that the key step of the formation of the target system is the electrophilic attack of the chromone ring followed by oxidative aromatization.

While optimizing conditions for a tree-component reaction for the synthesis of indole-substituted chromenopyridines, Kulikova et al. revealed that carrying out reactions of salicylic aldehydes **148** and *N*-methylpiperidone **149** in EtOH in the presence of *L*-proline as a catalyst leads to the formation of hexahydrochromenopyridines **150** in 69–76% yields (Figure 38) [58]. The products precipitated from the reaction mixtures and were isolated by filtration. According to the X-ray data, compounds **150** are formed diastereoselectively and have hydroxyl groups in a relative *trans*-configuration and the azadecalin system of annulated two non-aromatic six-membered cycles in a *cis*-configuration. It was demonstrated that *L*-proline played a crucial role in the stereoselectivity; thus, when the same reactions were performed without the additive, a series of chromeno[3,2-*c*]pyridines **151** with *trans*-configuration of the fused azadecalin system were obtained.

A simple and interesting way for the synthesis of chromeno[3,2-*c*]pyridines **154** was suggested by Kawai et al. The starting pyrano[4,3-*b*]chromen-1,10-dione, derived through acylation of pyran-2-one with 2-fluorobenzoyl chloride and the consequent rearrangement and aromatic nucleophilic substitution caused by the treatment of resultant ester **152** with KCN, Et_3_N, and 18-crown-6, reacted with primary amines **153** in the presence of acetic acid at 110 °C in trifluoroethanol to provide *N*-substituted chromeno[3,2-*c*]pyridines **154** in 16–32% yields (Figure 39) [88].

## 4. Biological Activity of Chromeno[3,2-*c*]Pyridines

Natural and synthetically prepared chromeno[3,2-*c*]pyridines demonstrate a wide range of biological activities. The available data is summarized in Table 1.

Comparing the biological activity of other chromenopyridines [89,90] with chromeno[3,2-*c*]pyridines clearly demonstrates that the latter has promising potential for treating neurodegenerative diseases [56,57,58,69]. At the same time, the antimicrobial and antibacterial activities of these compounds are similar to other chromenopyridines [91,92,93]. However, the antivirus activity was described only for chromeno[3,2-*c*]pyridines. Some chromenopyridines can be used as non-steroidal anti-inflammatory drugs [94,95,96]. Unfortunately, chromeno[3,2-*c*]pyridines have not been tested for these purposes yet.

Chromeno [2,3-*b*]pyridines and chromeno [4,3-*b*]pyridines possess high antiproliferative properties [97,98,99,100,101]. However, the anticancer properties of chromeno[3,2-*c*]pyridines do not look very promising now, but the development of their chemistry can produce novel substances, demonstrating better results in this field. For example, the introduction of indole fragments increases their potential.

In our opinion, we assume that chromeno[3,2-*c*]pyridine are at the beginning of their stardom, and the further evolution of the chemistry of chromeno[3,2-*c*]pyridines should be focused on the synthesis of multitarget compounds, having both anti-neurodegenerative and antioxidant properties. These aims may be achieved by the introduction of substituents that have described antioxidant activities. Another productive approach implies the synthesis of binary compounds containing two scaffolds chromeno[3,2-*c*]pyridine and, for example, tacrine, indole, etc., which are bonded by linkers [102,103].

**Table 1 molecules-29-04997-t001:** Information on biological activity of chromeno[3,2-*c*]pyridines.

Chemical Structure	Clinical Use	Concentration of Compound	Year–Author–Lit
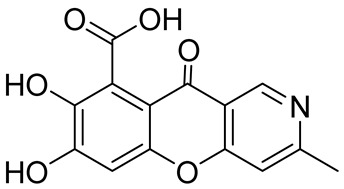 SB236049	inhibitory activity towards the Bacillus cereus II and Bacteroides fragilis CfiA metallo-β-lactamases	IC_50_ values of 0.3 μM and 2 μM	In 2002, Paune et al. [33] isolated from the fungus Chaetomium funicola
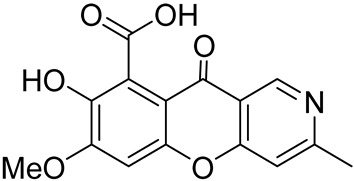	the alkaloid displayed inhibitory activity against New Delhi metallo-β-lactamase	IC_50_ value of 87.9 μM	In 2013, Gan et al. [34] isolated cultures of Penicillium sp. I09F 484
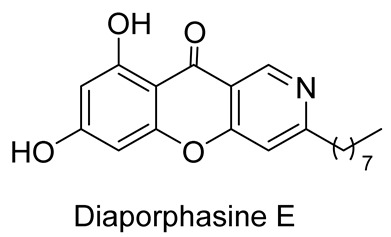	cytotoxicity against the tested cell lines L929 and KB3.1	IC_50_ values of 0.9 and 3.7 μM	In 2023, Phutthacharoen et al. [36] isolated by Phutthacharoen et al. from a mycelial extract of a saprotrophic fungus Lachnum sp. IW157 growing on the common reed grass Phragmites communis
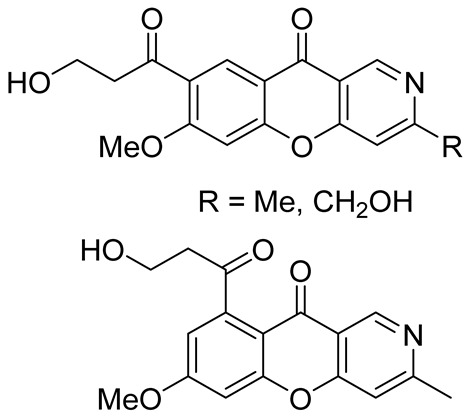	antirotavirus activity	TI values of 18.3, 23.7, and 19.2 therapeutic index(TI)—CC_50_/EC_50_. CC_50_: mean (50%) value of cytotoxic concentration; EC_50_: mean (50%) value of effective concentration	In 2021–2022, Yang et al. [40] isolated from plants of Thalictrum scabrifolium
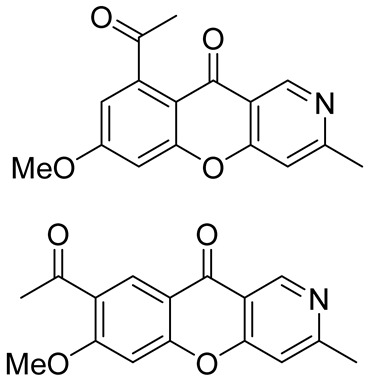	antibacterial activity against 12 microbial strains isolated from the saliva of smokers	antibacterial activity in the range of 11.1 to 35.3 mm	In 2022, Yin et al. [41] isolated from plants of Thalictrum scabrifolium
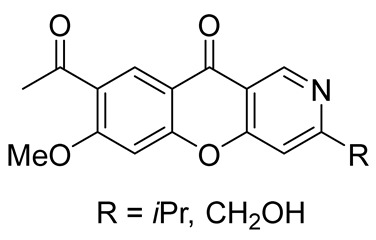	antirotavirus activity	TI values of 19.7 and 17.1 therapeutic index(TI)—CC_50_/EC_50_. CC_50_: mean (50%) value of cytotoxic concentration; EC_50_: mean (50%) value of effective concentration	In 2022, Hu et al. [42] isolated from plants of Thalictrum finetii
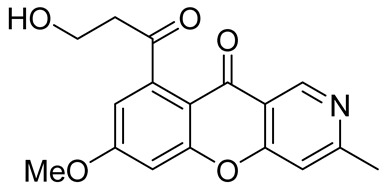	anti-tobacco mosaic virus (anti-TMV) activity	IC_50_ value of 29.3 μM	In 2023, Wu et al. [44] plants of Thalictrum microgynum
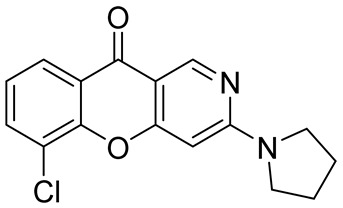	efficacy in significantly reducing chronic hepatitis B virus (HBV) antigens, DNA, and intrahepatic cccDNA levels	IC_50_ value of 0.51 μM	In 2022, Chen et al. [52]
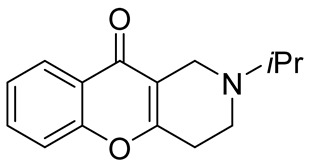	inhibitors of the human isoforms of MAO A	IC_50_ value of 4.4 μM	In 2020, Purgatorio et al. [57]
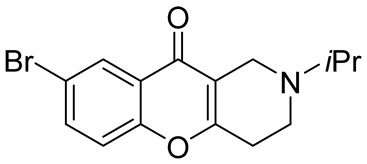	inhibitors of the human isoforms of MAO B and AChE.	IC_50_ values of 2.23 μM and 3.22 μM	In 2020, Purgatorio et al. [57]
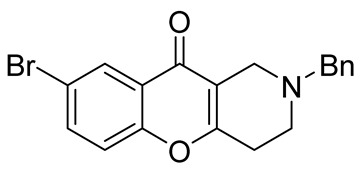	inhibitors of the human isoforms of MAO B and AChE.	IC_50_ values of 4.98 μM and 17.3 μM	In 2020, Purgatorio et al. [57]
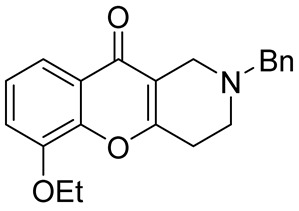	inhibitors of the human isoforms of MAO B	IC_50_ value of 0.89 μM	In 2020, Purgatorio et al. [57]
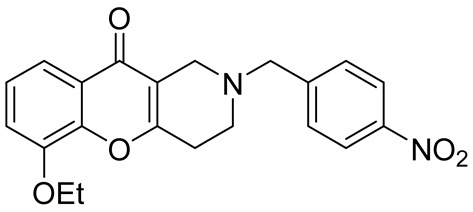	inhibitors of the human isoforms of MAO B and AChE	IC_50_ values of 1.15 μM and 23.0 μM	In 2020, Purgatorio et al. [57]
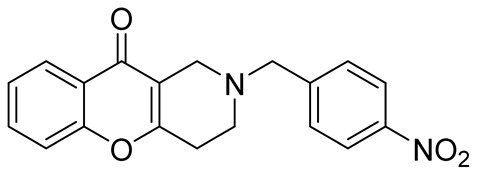	inhibitors of the human isoforms of MAO A, MAO B, and AChE	IC_50_ values of 7.1 μM, 2.08 μM and 3.43 μM	In 2020, Purgatorio et al. [57]
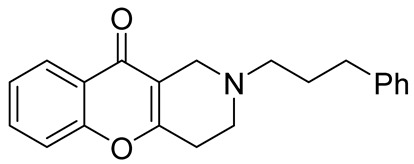	inhibitors of the human isoforms of MAO B and BChE	IC_50_ values of 2.81 μM and 3.87 μM	In 2020, Purgatorio et al. [57]
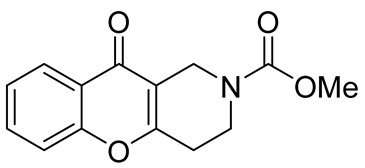	inhibitors of the human isoforms of MAO A, MAO B and AChE	IC_50_ values of 1.14 μM, 4.91 μM, and 2.05 μM	In 2020, Purgatorio et al. [57]
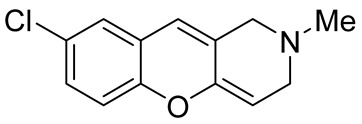	inhibitors of the human isoforms of MAO A	IC_50_ value of 1.18 μM	In 2023, Kulikova et al. [58]
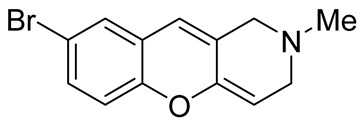	inhibitors of the human isoforms of MAO A and B	IC_50_ values of 0.703 μM and 7.88 μM	In 2023, Kulikova et al. [58]
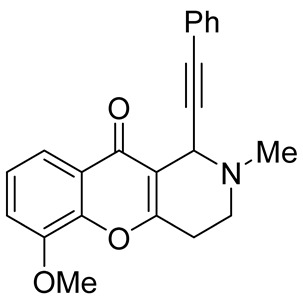	inhibitors of the human isoforms of MAO B	IC_50_ value of 0.626 μM	In 2023, Kulikova et al. [58]
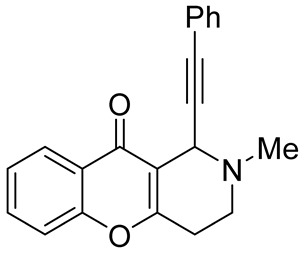	inhibitors of the human isoforms of MAO B and ChE (AChE and BChE).	IC_50_ values of 0.510 μM, 6.78 μM, and 4.42 μM	In 2023, Kulikova et al. [58]
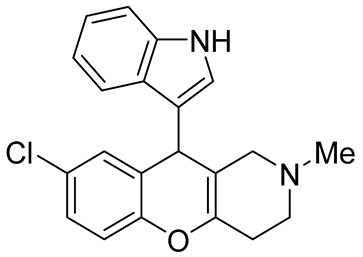	inhibitors of the human isoforms of MAO B the tumor growth inhibitory activity assayed in three cell lines (i.e., MCF-7, HCT116, and SK-OV-3)	IC_50_ value of 7.3 μM IC_50_ value of 4.8 μM (MCF-7) IC_50_ value of 8.62 μM (HCT116) IC_50_ value of 14.7 μM (SK-OV-3)	In 2023, Kulikova et al. [58]
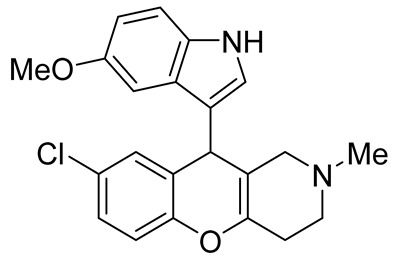	inhibitors of the human isoforms of MAO B the tumor growth inhibitory activity assayed in three cell lines (i.e., MCF-7, HCT116 and SK-OV-3)	IC_50_ value of 4.72 μM IC_50_ value of 6.62 μM (MCF-7) IC_50_ value of 18.6 μM (HCT116) IC_50_ value of 22.3 μM (SK-OV-3)	In 2023, Kulikova et al. [58]
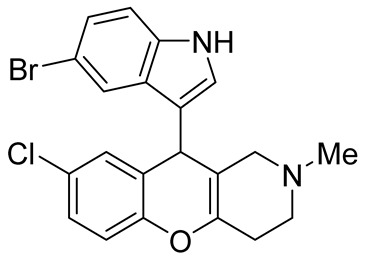	inhibitors of the human isoforms of MAO B the tumor growth inhibitory activity assayed in three cell lines (i.e., MCF-7, HCT116 and SK-OV-3)	IC_50_ value of 3.51 μM IC_50_ value of 4.83 μM (MCF-7) IC_50_ value of 9.40 μM (HCT116) IC_50_ value of 11.3 μM (SK-OV-3)	In 2023, Kulikova et al. [58]
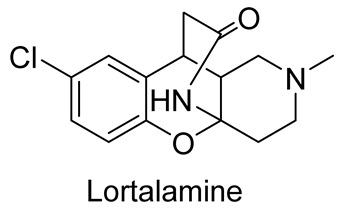	antidepressant activity	Lortalamine is a potent NET inhibitor with a potency higher than imipramine (13 fold) and desipramine (5 fold)	In 1980, Briet et al. [69], in 1985, Depin et al. [104], in 2006, Ding et al. [105]
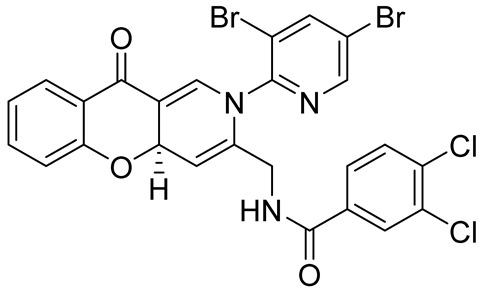	activity against Mycobacterium tuberculosis H37Rv (MTB) and multidrug-resistant M. tuberculosis (MDR-TB).	IC_50_ value of 50.69 μM	In 2010, Sriram et al. [80]

## 5. Conclusions

Chromeno[3,2-*c*]pyridines and their derivatives are promising heterocyclic systems due to their biological properties. We reviewed the literature, which reported various protocols for the efficient synthesis of chromeno[3,2-*c*]pyridine core. Synthetic methods for chromenopyridines based on the formation of both chromene and pyridine moiety are described. Various factors affecting the yield, temperature, substituents, and solvent effects are also discussed.

The analysis of biological data clearly demonstrates the great potential of chromeno[3,2-*c*]pyridine core for medicinal chemistry. We hope that this review will be useful for the synthesis of new biologically active compounds having antiviral, antibacterial, antituberculous, and cytotoxic activities, inhibitory activity against MAO, AChE, BChE, and anti-aggregation activity against β-amyloid.

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
