# Peer review of "Synthesis and Biological Activity of Chromeno[3,2-c]Pyridines"

_molecules, 2024, doi:10.3390/molecules29214997_

Round 1
Reviewer 1 Report
Comments and Suggestions for Authors
I really appreciate having the opportunity to review this manuscript. The paper is rather well prepared and seems to include all the synthetic methods and their natural occurrence of Chromeno[3,2-c]Pyridines throughout the work. In this review article report synthesis of many of this classes of compounds including cyclization of pyridyl(quinolyl) phenyl ethers, intramolecular Diels-Alder reactions, palladium-catalyzed intramolecular arylation etc. Also, the review touches upon the vipieties and structure of the mother nitrogen bases and biological activity of natural Chromeno [3, 2-c] pyridine alkaloids.
However, as a reviewer of the work, I would like the authors to do just within the scope of this work, the authors could consider adding the following additional headings or subheadings:
- This section include accomplied SAR studies on chromeno[3,2-c]pyridine and chromeno[3,2-c]quinoline derivatives, justifying structural elements contributing for biological activities of these compounds with special emphasis to antimicrobial, antiviral and cytotoxic activity.
- However, some of the reasons for these restrictions could be overcome through the addition of a specific pharmacological potential of what these compounds can look like, their purpose inм drug development, especially in the strategy against microbes, viruses and cancer.
3. In this review, it might be possible to tackle some synthetic challenges or limitations inherent to the reported methodologies and propose potential future directions towards improving the efficiency and versatility of the synthetic strategies to these heterocyclic systems.
4. While doing so, the review can include a detailed section on some of the in-silico studies where molecular docking, structure-based design of drugs, or Insilico screening can be employed to study the binding aspects and structure activity relationship of the said compounds.
Comments on the Quality of English LanguageMinor editing of English language required.
Author Response
We are very grateful for your supportive comments.
Comments 1:This section include accomplied SAR studies on chromeno[3,2-c]pyridine and chromeno[3,2-c]quinoline derivatives, justifying structural elements contributing for biological activities of these compounds with special emphasis to antimicrobial, antiviral and cytotoxic activity.
Response 1: Thank you for the recommendation, the corresponding data have been added as Table 1.
Comments 2: However, some of the reasons for these restrictions could be overcome through the addition of a specific pharmacological potential of what these compounds can look like, their purpose in drug development, especially in the strategy against microbes, viruses and cancer.
Response 2: The necessary discussion has been added.
Comment 3: In this review, it might be possible to tackle some synthetic challenges or limitations inherent to the reported methodologies and propose potential future directions towards improving the efficiency and versatility of the synthetic strategies to these heterocyclic systems.
Response 3: The necessary discussion has been added.
Comment 4: While doing so, the review can include a detailed section on some of the in-silico studies where molecular docking, structure-based design of drugs, or Insilico screening can be employed to study the binding aspects and structure activity relationship of the said compounds.
Response 4: We completely agree with the reviewer that this information will be useful for biological purposes. However, our review is focused on synthetic strategies for the synthesis of target compounds and we would not like add information concerning in-silico studies.
Reviewer 2 Report
Comments and Suggestions for Authors
This review presents a highly valuable contribution to the fields of organic chemistry and medicinal chemistry by summarizing the synthesis and biological properties of chromeno[3,2-c]pyridines and chromeno[3,2-c]quinolines. This review is positioned to guide future developments in the synthesis of biologically active heterocycles with potential applications in drug discovery and materials science. I would like to recommend this review to be published after the authors address my following concerns:
- Please add a comparative section that contrasts the biological and synthetic profiles of chromeno[3,2-c]pyridines and chromeno[3,2-c]quinolines with other related chromenopyridine scaffolds, which will provide a broader context for their significance.
- A more structured presentation of the biological data (in tables) that summarizes the activities (antiviral, antibacterial, etc.) along with their effective concentrations (IC50 or MIC values) would make it easier for readers to navigate the biological significance of these compounds.
- Please provide a discussion on the potential of other related applications of Chromeno[3,2-c]Pyridines and a clear outline of future research direction.
Minor editing of English language is needed.
Author Response
We are very grateful for your supportive comments.
Comments 1: Please add a comparative section that contrasts the biological and synthetic profiles of chromeno[3,2-c]pyridines and chromeno[3,2-c]quinolines with other related chromenopyridine scaffolds, which will provide a broader context for their significance.
Response 1: The necessary information has been added.
Comments 2: A more structured presentation of the biological data (in tables) that summarizes the activities (antiviral, antibacterial, etc.) along with their effective concentrations (IC50 or MIC values) would make it easier for readers to navigate the biological significance of these compounds.
Response 2: Thank you for the recommendation, the corresponding data have been added as Table 1.
Comment 3: Please provide a discussion on the potential of other related applications of Chromeno[3,2-c]Pyridines and a clear outline of future research direction.
Response 3: The necessary discussion has been added.